# Quantitative and Qualitative Evaluation of Plant Intake in Laying Hens: n-Alkanes as Predictive Fecal Markers for Dietary Composition Assessment

**DOI:** 10.3390/ani14030378

**Published:** 2024-01-24

**Authors:** Laid Dardabou, José Carlos Martínez-Ávila, Markus Werner Schmidt, Károly Dublecz, Christiane Schwarz, Miguel Angel Ibáñez, Martin Gierus

**Affiliations:** 1Institute of Animal Nutrition, Livestock Products, and Nutrition Physiology, University of Natural Resources and Life Sciences, 1190 Vienna, Austria; laid.dardabou@boku.ac.at (L.D.); markus.schmidt@boku.ac.at (M.W.S.); christiane.schwarz@boku.ac.at (C.S.); 2Departamento de Economía Agraria, Estadística y Gestión de Empresas, Universidad Politécnica de Madrid, 28040 Madrid, Spain; jc.martinez.avila@upm.es (J.C.M.-Á.); miguel.ibanez@upm.es (M.A.I.); 3Institute of Physiology and Nutrition, Georgikon Campus, Hungarian University of Agriculture and Life Sciences, 2100 Keszthely, Hungary; dublecz.karoly@uni-mate.hu

**Keywords:** n-alkanes, laying hens, recovery rate, alfalfa, outdoor consumption, free-range

## Abstract

**Simple Summary:**

Providing better living conditions for laying hens by promoting free-range systems has sparked curiosity about potential alterations in their dietary habits. Understanding these changes can lead to new feeding strategies to meet their nutritional needs. However, this task is challenging due to the inherent complexities of the outdoor environments and variations in plant material breakdown. Our research marks a pioneering step in exploring both qualitative and quantitative methods for assessing plant intake in laying hens using n-alkanes as markers, and demonstrates their viability even at levels of plant material in the diet as low as 1%. It also highlights the need for further research on different sources of n-alkanes, especially in real case studies with different plant and insect species under outdoor conditions. Such studies are essential to provide a deeper understanding of the dietary factors affecting the performance of laying hens in free-range systems.

**Abstract:**

The shift in animal welfare standards towards free-range housing for laying hens in the EU has raised questions about changes in dietary composition. Accurate assessment of outdoor plant material intake is crucial for effective feeding strategies. This study introduces an approach using n-alkanes as markers to determine plant intake in laying hens, involving n-alkane recovery rate assessment, discriminant analysis and linear equation-solving for both qualitative and quantitative assessment, respectively, considering systematic n-alkane combinations. Two diets: a standard commercial diet and a diet incorporating 1% alfalfa were tested. Chemical analyses showed an altered n-alkane profile due to alfalfa inclusion, resulting a recovery rates ranging from 30–44% depending on the n-alkane type and diet. Statistical analysis revealed significant differences in recovery rates among the different alkanes for the same diets and between the diets for the same alkane, together with an interaction between n-alkane carbon chain length and initial concentration in the diet. The method accurately predicted plant inclusion, with a slight overestimation (2.80%) using the combination C25-C29-C33. Accurate qualitative classification of the animals based on fecal n-alkanes profiles was observed. The study successfully demonstrated the utility of n-alkanes for estimating dietary composition, providing a non-invasive approach for future free-range studies.

## 1. Introduction

In recent years, the European Union (EU) has witnessed a significant shift in animal welfare standards, with an increasing emphasis on free-range housing systems for laying hens. In Austria, about 20% of egg production is being generated from free-range production systems [1]. This transition from conventional cages to more spacious and humane environments has raised questions about potential changes in the dietary composition of these birds. Understanding and quantifying the intake of outdoor plant material by free-range laying hens is critical, as it can provide valuable insight for producers seeking to implement effective feeding strategies that align with the birds’ nutritional requirements. However, this attempt is challenging due to the inherent heterogeneity of outdoor environments and variations in the decomposition of plant material among different animal and plant species [2,3]. Identifying key plant species consumed outdoors and applying knowledge of asynchronous fluctuations in animal nutritional requirements arising from these outdoor foraging activities can significantly impact the management of these flocks [3,4]. Former studies [5] estimated that approximately 10% to 15% and 20% to 25% of total feed intake in broilers and laying hens, respectively, may come from pasture, highlighting the need for accurate assessment methods. Unlike feed, free-range birds’ plant intake cannot be controlled under practical circumstances and may be subject to variation [6,7].

Early attempts at free-range nutritional contribution evaluation involved field observations and direct monitoring of animals, but these methods proved time-consuming, required expertise in plant identification, and were often limited to observing one animal at a time [8]. Surgical techniques like esophageal surgeries and rumen fistulation were developed in ruminant nutrition, but similar procedures proved intricate and unsuitable for small animals, particularly non-ruminant species [3] and poultry [5]. Additionally, physiological processes sometimes rendered plant material unrecognizable [2]. As an alternative, non-invasive post-digestive approaches were explored, including micro-histological analysis of fecal cuticles. While offering advantages, this method demanded substantial labor, time, specialized field observers, and faced the challenge of non-proportional representation of plant material in feces [2,3,9].

The use of markers in the diet has been confirmed as a successful alternative, among which are n-alkanes, which are naturally found in the epicuticular waxes of plants as mixtures of differing carbon chain lengths. The n-alkanes have received considerable attention as fecal markers for estimating the diet composition of housed [10] and grazing animals [11], considering the unique and different alkanes patterns of each plant species and special plant organs [12,13,14,15]. While n-alkanes have proven valuable in estimating herbage intake in grazing ruminants, their application in poultry remains limited [16,17,18,19,20]. The main issue at hand is the lack of knowledge about recovery rates of n-alkanes in non-ruminant species, especially poultry, and the various factors influencing this parameter.

The current study aimed to assess the accuracy of the determination method for plant intake by laying hens using n-alkanes as markers, while evaluating the recovery rates of the various n-alkanes in feces. Estimations of plant inclusion into the diet were conducted through a linear equation-solving method that leveraged the non-negative least squares procedure. We hypothesized that changes in the dietary composition, resulting from external plant consumption, would affect the n-alkane profile in feces and consequently its recovery rate when compared to a diet consisting exclusively of commercial feed.

## 2. Materials and Methods

### 2.1. Husbandry and Diets

An individual caging trial was conducted at the Institute of Physiology and Nutrition, Georgikon Campus, Hungarian University of Agriculture and Life Sciences (Keszthely, Hungary). The research protocol of the trial was approved by the Animal Ethics Committee of the Institute of Physiology and Nutrition, Georgikon Campus, Hungarian University of Agriculture and Life Sciences (Keszthely, Hungary). The Ethics Committee approved the specific protocol in its decision: MÁB-3/2023. Forty-eight TETRA-SL pullets from a commercial flock at 18 weeks of age were randomly selected and individually weighed (1426 ± 17.2 g) before being placed in enriched cages (width: 26 cm, height: 45 cm; length: 45 cm). Barn temperature and lighting were maintained at 24 ± 3 °C and 16 h light/d, respectively, following the Bábolna Tetra-SL Commercial Layer Management Guide. The hens had *ad libitum* access to water and were fed a restricted diet (80 g/bird/day) in finely ground mash to prevent selectivity. The experiment was conducted in 2 phases: a 5-day adaptation phase and a 2-day period for total fecal collections. Hens were fed two types of diets: a standard diet based formulated in line with the breeder’s recommendations for Tetra-SL [21] and a diet incorporating 1% of Alfalfa (*Medicago sativa*; dried, developmental stage according to mean stage by count: 4 to 6 (>3 nodes with buds to >2 nodes with open flowers [22]), ground to 1 mm), at the expense of dry feed (*w*/*w*). The choice of alfalfa in this study was due to its well-documented status as one of the most extensively researched plants in the context of animal nutrition, particularly in the case of ruminants, and its high availability in outdoor areas in Austria. Furthermore, the decision to incorporate alfalfa into the mixed diet at a rate of 1% was chosen to correspond to approximately 5% of the minimum contribution from outdoor sources to the diet of laying hens, as reported previously [5]. By opting for a 1% inclusion rate, the study aimed to challenge the model under conditions that align with practical scenarios, thereby enhancing the relevance and applicability of the research findings to poultry nutrition and management practices. In this regard, alfalfa plant was a mixture of leaves and stems. Detailed ingredient composition and nutrient content of the commercial diet are outlined in Table 1. Daily feed intake and total excreta output for each bird were meticulously recorded throughout the experiment. Total individual excreta (2 sampling days from each animal) were homogenized, weighed, and freeze-dried before analysis using VirTis AdVantage Plus freeze dryer Model XL-70.

### 2.2. Proximal Composition and Chemical Analyses

Feed samples were analyzed for moisture by oven-drying (DM; no. 3.1), ash with a muffle furnace (CA; no. 8.1), and N by Kjeldahl (CP; no. 4.1.1) as described by [23]. Crude protein was calculated as N × 6.25. The gross energy (GE) content was performed using an adiabatic bomb calorimetry (IKA C 200, IKA Werke GmbH & Co., KG, Staufen, Germany).

### 2.3. n-Alkane Analysis

n-Alkane concentration in feed, feces, and plant material samples was determined using a modification of [24] methodology and gas chromatography. Accurately weighed duplicates of 1 g dried and finely ground (1mm) samples were placed in a screw-top Pyrex tube (200 × 20 mm) before the addition of 0.15 g of internal standard [n-Tetracosane (C24) and n-tetratriacontane (C34) solved in Undecane] and 10 mL of 1.5 M KOH. Tube contents were mixed thoroughly and placed in a water bath (90 °C) for 4.5 h with continuous shaking. After cooling of samples to 70 °C by adding ice to the water bath to achieve partial cooling from 90° to 70 °C, 8.0 mL of heptane and 5.0 mL of distilled water were added to the contents, mixed thoroughly and placed again on the water bath until just boiling. The top organic phase was removed with a Pasteur pipette and transferred to a Pyrex 100 × 18 mm tube, and the process was repeated with 5mL of heptane. The contents were evaporated to dryness using Reacti-Therm III TS-18824 Heating Module in combination with nitrogen blowing on the surface to avoid oxidation, re-dissolved in 2 mL of heptane, and applied to an SPE-column (Discovery^®^ DSC-Si SPE Tube [500 mg, 3 mL] from Supelco Inc., Bellefonte, PA, USA) consisting of silica gel contained in disposable pipette tips stoppered with glass wool. The column was eluted with 8 mL of heptane in four steps of 2 mL each. The eluent was again evaporated to dryness and re-dissolved in 1 mL of heptane before injection into the capillary column in a gas chromatograph. The separation of n-alkanes was performed on an Agilent 78902A (Agilent Technologies, Santa Clara, CA, USA) gaschromatography module with an Agilent 7693B automatic liquid sampler supported by a multimode inlet (split/splitless), using a capillary column (Restek Rtx-1 30 m, 0.53 mm ID, 0.25 µm film thickness, with an upstreamed 5 m Integra-Guard column). GC settings were as follows: Injector: temperature 300 °C, gas pressure: 1.4774 psi, column: 41.689 mL/min gas flow, 1 µL injection volume, splitless); oven: temperature program: Initial: 100 °C, hold for 1 min, ramping 8 °C per minute to 310 °C, hold for 12.75 min; total run time: 40 min; FI-Detector Settings: temperature: 320 °C, H_2_ flow: 35 mL/min, air flow (synthetic air): 400 mL/min, makeup flow (N_2_): 15 mL/min. The following major (odd-chain) n-alkanes were analyzed: C25, C27, C29, C31, and C33.

### 2.4. Calculations and Statistical Analyses

The recovery rate of n-alkanes obtained in each animal was analyzed using a two-way factorial ANOVA considering diet and type of n-alkane as factors. Mean estimation and comparison were conducted using the emmeans package [25], and figures were generated using the ggplot2 package [26]. In order to ensure ANOVA assumptions of normality and homogeneity were met, residual plots were examined.

Qualitative assessment of the intake based on the n-alkanes profiles in feces was performed. Discriminant analysis was chosen as a pivotal statistical tool, and the analyses were conducted both collectively and individually for each n-alkane using the MASS package [27], aiming to determine the individual contributions in distinguishing between the groups of diet. In this study, all statistical analyses were conducted using the R programming system [28].

Furthermore, for quantitative evaluation, and based on the qualitative assessment results, the estimation of plant consumption involved the use of both dietary components (feed and alfalfa) and n-alkane profiles of excreta. Prior to this estimation, adjustments were made to n-alkane concentrations in excreta, considering the specific n-alkane recovery rate for each diet. The estimates were made using the non-negative least squares procedure proposed by [29]. For each experimental animal, the following system of linear equations was proposed: (1)$$x_{f}F_{i}+x_{a}A_{i}=E_{i}$$
where F_i_, A_i_ and E_i_, i = 25, 27, 29, 31, 33, are the concentrations of the n-alkane number i in feed, alfalfa and excreta, respectively.

The coefficients x_f_ and x_a_ were determined while imposing a non-negativity constraint on the parameter estimates. The nnls package [30], an implementation of the non-negative least squares algorithm of the R program system [28], was used for this purpose.

Once the coefficients x_a_ and x_f_ were estimated, the proportion of the dietary DM supply from plants could be calculated as the ratio: (2)$$x_{a}/\left(x_{a}+x_{f}\right)$$

The predictive ability of the method was assessed by calculating the mean square of error (MSE) made when predicting the consumption of the animals within each diet. The MSE was decomposed into the variance and squared bias components.

Besides solving the designed equation including all the five studied n-alkanes to identify the plant intake, the proposed equation was systematically solved using all possible combinations of the five n-alkanes, ranging from two to five. This comprehensive approach guarantees the selection of the best n-alkane combination with a high level of confidence in its predictive accuracy and reliability.

## 3. Results

The inclusion of 1% alfalfa in the diets had no discernible impact on the analyzed chemical composition of the diets and consequently did not affect the daily feed intake, which remained at approximately 70 g/day per animal, as shown in Table 2.

Due to the high concentrations of n-alkanes in the epicuticular wax of the upper part of alfalfa plants, especially C29 and C31, the addition of 1% alfalfa to the commercial diet markedly altered the n-alkane profile of the final mixture (Figure 1). In particular, an increase of 8%, 38%, 120%, 186%, and 27% was observed for C25, C27, C29, C31, and C33, respectively, compared to the n-alkane profile of the commercial diet (Table 2).

The different n-alkane profiles in the diets are also reflected in the concentration of n-alkanes in the excreta, especially in the n-alkanes C29 (2.87 vs. 7.03 mg/kg DM) and C31 (1.97 vs. 7.45 mg/kg DM) with much higher concentrations in the mixed than in the commercial diet (Figure 2).

Approximately 36% of the total n-alkanes assayed were indigestible in the commercial diet, whereas this figure increased to 41% with the inclusion of alfalfa in the diet (Table 3). One animal in the control group was identified as an outlier with a very high recovery rate (66% for the total alkanes) due to relatively low feed intake (42 g/day), and was therefore disregarded in the subsequent analysis. Table 3 provides the estimated means of the recovery rates of n-alkanes in each diet. The interaction between diet and type of n-alkane was significant (*p* value < 0.001), indicating that the mean differences between the n-alkanes depend on the type of diet consumed by the animal. Although the mean recovery rate was always higher in the diet with 1% alfalfa than in the diet without alfalfa, the differences were greater in the n-alkanes with greater carbon length (C29, C31, C33). In the commercial feed diet, the average recovery rate decreased as the carbon chain length increased until the minimum was obtained at the n-alkane C31. However, in the mixed feed diet the minimum was reached in the n-alkanes C27 and C29 and increased for the n-alkanes C31 and C33. These estimated mean values were used for the adjustments made to the concentrations of n-alkanes in excreta necessary for the application of the alfalfa consumption prediction.

The results of qualitative evaluation using the discriminant analysis, as shown in Table 4, demonstrate the effectiveness of both the comprehensive use of all n-alkanes and individual n-alkanes in classifying animals based on their diets as reflected in their fecal n-alkane profiles. Using all n-alkanes together results in a high degree of accuracy in grouping animals with similar diets. However, when each n-alkane is considered individually, C29 and C31 consistently provide accurate results. In contrast, C25 shows some discrepancies, misclassifying eight animals from the mixed diet as part of the control group and seven animals from the control diet as part of the mixed-diet group. Conversely, C27 and C33 demonstrate lower levels of error, with only one control-diet animal misclassified as part of the mixed-diet group and two mixed-diet animals misclassified as part of the control group. These findings have led to an investigation to identify the most accurate combination of the n-alkanes studied under the proposed equation, using all possible combinations to optimize accuracy.

Figure 3 presents notable correlations observed between various n-alkanes in feces, where strong associations were identified. In particular, the correlation between C31 and C29 showed a Pearson correlation coefficient (r) value of 0.98, indicating a strong relationship. Additionally, C31 and C33 presented a high r of 0.92, while the correlation between C29 and C33 provided an r of 0.90, indicating substantial associations between these pairs of n-alkanes. Furthermore, the correlations between C31 and C27, as well as C27 and C29, demonstrated substantial links with r values of 0.82 and 0.86, respectively, further highlighting the strength of these associations.

Table 5 presents key evaluation metrics, including root mean square error (RMSE), variance and bias, used to assess the accuracy of predictions regarding alfalfa consumption in animals fed the two experimental diets. In particular, the coefficient x_a_ was close to zero (x_a_ = 0.001) for all animals assigned to the commercial diet, indicating that the method effectively predicted the absence of alfalfa in this particular diet, with an average alfalfa inclusion rate of approximately 0.02%. When considering animals on the mixed diet, the variance and bias were calculated to be 0.012 and −0.196, respectively. The mean prediction of the proportion of alfalfa in the diet stood at 0.804, revealing a moderate underestimation of 0.196. The RMSE, quantified at 0.224, represents an error equivalent to 28% of the mean (0.224/0.804). Importantly, the overall application of the entire range of alkanes examined in solving the equation showed a relatively accurate prediction of the proportion of alfalfa added to the diet on a dry matter basis.

Table 6 provides an overview of the outcomes resulting from a systematic evaluation of various combinations of the n-alkanes studied to solve the equation, showing the best solutions based on the minimum error committed. In the context of a five-component combination, it is noteworthy that an underestimation of 19.6% was observed, with the estimated mean value of alfalfa incorporated in the diet at approximately 0.804%. As for the four-component combination, the results showed that the most accurate solution was achieved using the combination C25–C27–C29–C33, which had a minimum committed error of 3.5% and an estimated mean value of 0.965% for alfalfa inclusion in the diet. However, the highest error was associated with the combination C27–C29–C31–C33, resulting in a significant bias of 24.4% (Appendix A Table A1). Among the three-component combinations, the optimal accuracy was associated with the combination C25–C29–C33, which resulted in a slight overestimation of 2.80% with an estimated mean of 1.028% for alfalfa inclusion in the diet. On the other hand, the combination C25–C27–C33 showed a maximum bias of about 41.7%. Finally, the two-component combinations demonstrated a minimum committed error with a slight overestimation of 4.40% when using the combination C25–C29, giving an estimated mean of 1.044% for alfalfa in the diet. The largest error in this category, and in general, was found in the combination C27–C33, which estimated the inclusion of alfalfa in the diet at about 14.25%.

## 4. Discussion

n-Alkane recovery in feces is generally incomplete, and is known to increase, in a predictable way, with carbon-chain length of individual n-alkanes [24]. The recovery rates of n-alkanes (C27–C33) in this study varied from 0.30 to 0.44 (Table 3), dependently of the diet and the carbon chain length of these n-alkanes, as seen in the significant interaction (*p*-value < 0.05), where recovery rate was higher in alfalfa including diet. These findings are consistent with the published report by [19]. However, higher recovery values were reported by [16] and utilized by [20], where the range was 0.55 to 0.75 for C27 and C33, respectively. This substantial difference can be attributed to the higher concentration of n-alkanes presents in the diets used by [16], where a grass-fed treatment was administered via tube feeding, with the examined dried grass containing 12.0, 31.5, 131.2, 168.2, 78.5 mg/kg DM of C25, C27, C29, C31, and C33, respectively, in comparison to 1.28, 2.20, 5.69, 5.13, 1.51 mg/kg DM tested in the mixed diet as part of the current study. The high concentration of n-alkanes in plants is primarily associated with C29 and C31, measuring at 289 and 358 (mg/kg DM) in the alfalfa batch used (Table 2), respectively, exhibiting a notably favorable recovery rate. This observation aligns with the results reported by both [16,19], which highlights the potential of these two n-alkanes as reliable markers compared to the rest.

The applied methodology demonstrated favorable outcomes despite a relatively low plant inclusion rate of 1% in the diet, contrasting with the findings of Rivera [31] in a field-based case study on broilers where plant inclusion was approximately 10% of the daily intake (11 g DM herbage per 100 g DM feed). However, this effect might be significant in laying hens, as they can consume as much as 30–40 g of dry matter (DM) from a grass/clover pasture per 110 g DM of feed daily [7,32,33]. This difference represents a ten-fold higher inclusion level. Furthermore, this increased plant inclusion rate can minimize potential errors due to the low concentration of n-alkanes in the experimental diet used in the current study.

The investigated method moderately underestimated the proportion of alfalfa (bias = −0.196) when using the entire set of the studied n-alkanes. This underestimation was attributed to the higher variability observed, where approximately 10 g of daily leftovers per animal accounted for approximately 13% of the diet offered daily. Given that the planned inclusion of alfalfa was 1%, this indicates a potential for lower alfalfa consumption compared to the leftovers if animals showed selectivity in their feeding behavior. However, despite this observed discrepancy, the methodology demonstrated notably high accuracy and did exhibit a very low error in determining whether the animals consumed only one source of n-alkanes, such as the commercial diet. In a practical context, it is important to consider that other sources of n-alkanes, such as those from soil, as reported by [19], could be part of the diet studied. Therefore, the additional inclusion of n-alkanes from sources such as soil may lead to an underestimation of plant consumption. In order to address this, the inclusion of soil n-alkane profiling in the study area may be valuable to improve consumption estimates.

To the best of our knowledge, this research marks the first investigation of both qualitative classifications based on fecal n-alkane profiles using discriminant analysis and the role of various n-alkane combinations in quantitatively estimating plant intake in poultry. Regarding the qualitative assessment of the diet, the proposed methodology demonstrated exceptional accuracy in classifying the type of diet based on n-alkane concentrations, while using individual n-alkanes led to some classification errors. The best results emerged when using the combination of three n-alkanes (C25, C29, C33), whereas the least precise predictions were obtained when employing all five studied n-alkanes. However, it is crucial to acknowledge the impact of multicollinearity (Appendix A), particularly among C31, C29, and C27, when used together, which can contribute to higher prediction errors. Eliminating C31 from the n-alkanes used for solving the equation led to an 82% improvement in the prediction error, significantly reducing the underestimation. A slight improvement in prediction error was observed when C27 was removed, resulting in a 22% reduction in the error compared to using C25, C27, C29, and C33 together. Even when combining only two n-alkanes (C25 and C29), the prediction error increased slightly, but it remained significantly lower than when using all five n-alkanes together. It is noteworthy that our results deviate from the conclusions reached by [34] and Dove and [35], where the use of more n-alkanes was associated with better predictive precision. These findings highlight the critical role of n-alkane selection in predictive accuracy.

## 5. Conclusions

In conclusion, the use of fecal n-alkane profiles for the qualitative grouping of animals based on intake type has proven to be a successful and valuable approach. Similarly, the use of n-alkanes for quantitative estimation of plant intake in laying hens has shown promise, offering the advantage of avoiding the need for artificial dosing with external markers. Furthermore, this method offers the potential to not only estimate diet composition, but also to break down intake into individual n-alkane sources, which holds significant value for future free-range research. In order to achieve the highest predictive accuracy, careful selection of optimal n-alkane combinations is of major importance. However, refinement is needed to accurately estimate the contribution of n-alkanes from other sources, which requires in-depth analysis of the initial dietary n-alkane profile and its impact on the recovery rate. Furthermore, it is imperative that future studies explore additional sources of n-alkanes, particularly in the context of different plant and insect species under outdoor conditions. Such efforts are essential to achieve a more accurate estimation of plant intake in real-world scenarios, thereby improving our understanding of the dietary factors that influence the performance of laying hens in a free-range system.

## Figures and Tables

**Figure 1 animals-14-00378-f001:**
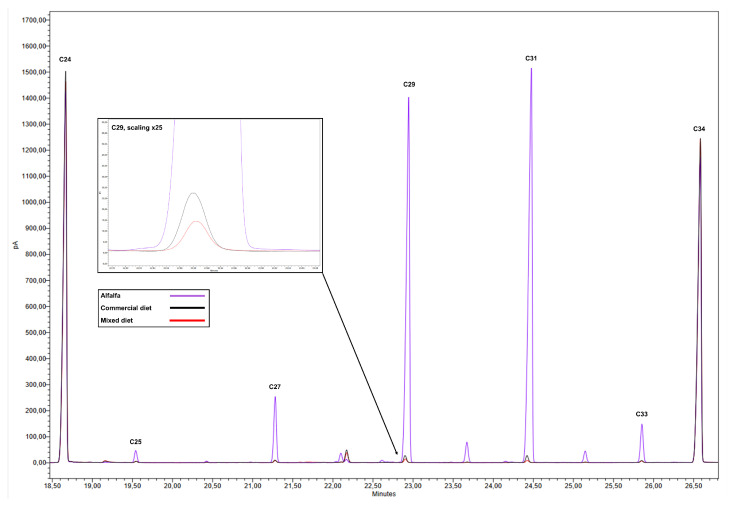
The GC/MS profile of alkane patterns for diet, alfalfa and feces used to estimate alkane concentrations of intake components.

**Figure 2 animals-14-00378-f002:**
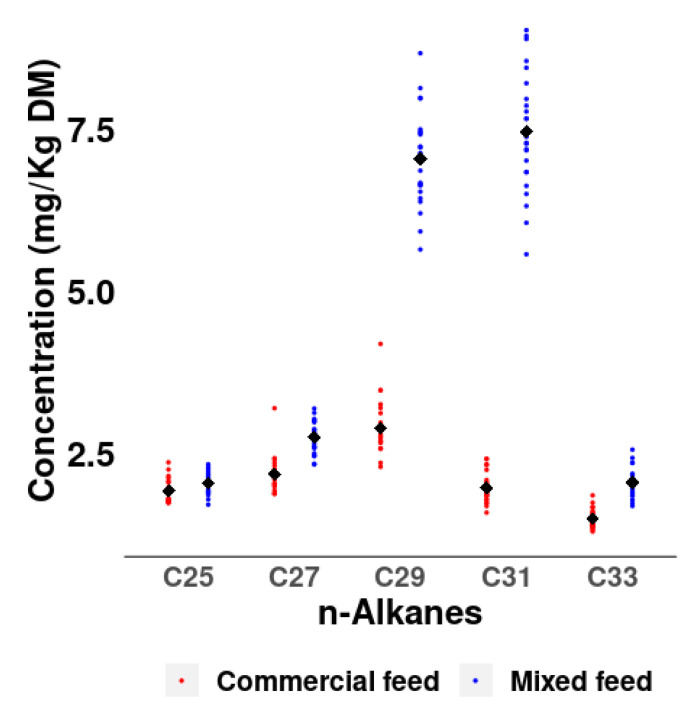
n-Alkane concentrations in excreta of hens fed with commercial and mixed feed (with 1% alfalfa inclusion).

**Figure 3 animals-14-00378-f003:**
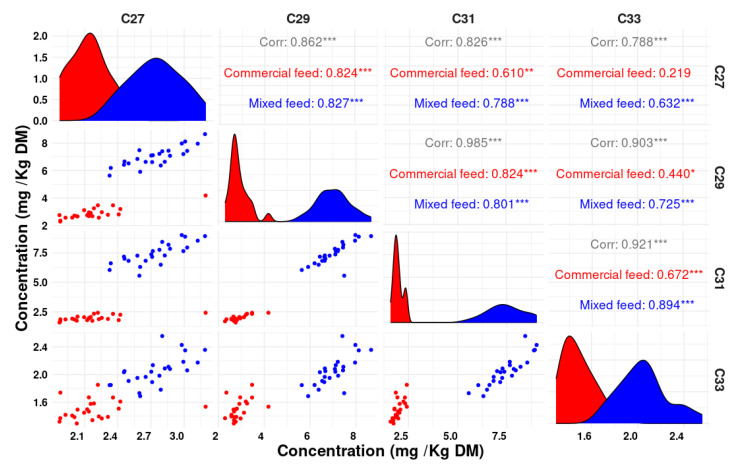
Scatterplot of n-alkane contents in the feces of laying hens fed the control diet (red) and the mixed diet (blue (* *p* ≤ 0.05, ** *p*≤ 0.01, and *** *p*≤ 0.001).

**Table 1 animals-14-00378-t001:** Ingredient composition (as fed basis) and calculated nutrient content of the commercial diet.

Composition of the Diets	%
Corn	28.6
Wheat	42.0
Extracted soybean meal	8.80
Sunflower meal	8.50
DDGS	3.00
Wheat feed flour	3.00
Sunflower oil	1.50
Limestone 2–4 mm	1.00
Monocalcium phosphate	0.65
Limestone grit	0.61
Premix	0.50
NaCl	0.26
L-lysine 78%	0.11
L-methionine 99%	0.07
Phytase	0.528
NSP-degrading enzymes	0.83
Calculated nutrient content	
Dry matter	88.6
Ash	5.52
Crude protein	15.6
Ether extract	3.59
Crude fibre	4.72
Ca	0.88
P (total)	0.55
Na	0.14
Lysine	0.69
Methionine	0.34

**Table 2 animals-14-00378-t002:** Analytical characteristics and n-alkanes concentrations of alfalfa (*Medicago sativa*), commercial and mixed feeds (/kg DM).

	Alfalfa	Commercial Feed	Mixed Feed (with 1% Alfalfa Inclusion)
Daily feed intake ^1^ (g)		70.3	68.6
Dry matter (g/kg)	906	890	891
Crude protein (g/kg)	206	161	163
Ash (g/kg)	90.3	40.6	41.1
Gross energy (MJ)	18.5	18.6	18.6
n-Alkanes (mg)			
C25	8.12	1.18	1.28
C27	44.3	1.59	2.20
C29	289	2.58	5.69
C31	358	1.79	5.13
C33	26.9	1.19	1.51
n-Alkanes (C25–C33)	719	7.15	14.5

^1^ Alfalfa incorporation effect was non-significant for daily feed intake (*p* > 0.05).

**Table 3 animals-14-00378-t003:** Estimated means, standard errors and *p* values from two-way ANOVA of fecal recovery rate for diet and n-alkane types as factors.

**Diet**	**Commercial Feed**						**Mixed Feed (with 1% Alfalfa Inclusion)**						***p* Value**		
**n-Alkanes**	**C25**	**C27**	**C29**	**C31**	**C33**		** C25**	**C27**	**C29**	**C31**	**C33**		**Diet**	**n-Alkanes**	**Interaction**
Estimated mean ^1^	0.439 ^aB^	0.369 ^bB^	0.302 ^cB^	0.297 ^cB^	0.339 ^bB^		0.479 ^aA^	0.375 ^cdB^	0.372 ^dA^	0.437 ^bA^	0.409 ^bcA^		<0.001	<0.001	<0.001
SEM ^2^	0.0094						0.0092								

^1^ n = 23 for control and 24 for mixed diet with 1% alfalfa. ^^a, b, c, d^^ Means within each diet lacking a common superscript differ significantly (*p* < 0.05). ^A,B^ Means within each n-alkane lacking a common superscript differ significantly (*p* < 0.05). ^2^ SEM Standard error of the mean.

**Table 4 animals-14-00378-t004:** Discriminant analysis of fecal n-alkanes for laying hens diet classification.

				**Commercial diet GROUP (n = 23)**			**Mixed Feed Diet Group (n = 24)**	
	**n-Alkane**	**LD1 ^1^**		**Commercial Diet**	**Mixed Feed Diet (with 1% Alfalfa Inclusion)**		**Commercial Diet**	**Mixed Feed Diet (with 1% Alfalfa Inclusion)**
n-Alkanes (C25–C33)	C25	−0.10		23	0		0	24
	C27	−1.20						
	C29	3.32						
	C31	5.77						
	C33	−1.90						
Individual	C25	1.05		16	7		8	16
	C27	1.50		22	1		2	22
	C29	3.67		23	0		0	24
	C31	4.19		23	0		0	24
	C33	1.82		22	1		2	22

^1^ Coefficients of first linear discriminant function.

**Table 5 animals-14-00378-t005:** Prediction of the proportion of alfalfa in total dry matter intake (%) in hens fed a diet with 1% alfalfa using all the studied n-alkanes.

	Mean ^1^	RMSE ^2^	Bias	Var ^3^
Commercial diet	0.017	0.032	0.017	0.001
Mixed feed diet	0.804	0.224	−0.196	0.012

^1^ Mean of prediction (n = 24). ^2^ RMSE root mean squared error of prediction. ^3^ Var variance of the error of prediction.

**Table 6 animals-14-00378-t006:** Best possible combinations of n-alkanes to predict the proportion of alfalfa in total dry matter intake (%) in hens fed a diet with 1% alfalfa.

	Mean ^1^	RMSE ^2^	Bias	Var ^3^
C25–C27–C29–C31–C33	0.804	0.224	−0.196	0.012
C25–C27–C29–C33	0.965	0.133	−0.035	0.016
C25–C29–C33	1.028	0.124	0.028	0.015
C25–C29	1.044	0.150	0.044	0.020

^1^ Mean of prediction (n = 24). ^2^ RMSE: root mean squared error of prediction. ^3^ Var: variance of the error of prediction.

## Data Availability

The data presented in this study are available in article.

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
