# Peer review of "Quantitative and Qualitative Evaluation of Plant Intake in Laying Hens: n-Alkanes as Predictive Fecal Markers for Dietary Composition Assessment"

_animals, 2024, doi:10.3390/ani14030378_

Round 1
Reviewer 1 Report
Comments and Suggestions for Authors
The manuscript Quantitative and Qualitative Evaluation of Plant Intake in Laying Hens: n-Alkanes as Predictive Fecal Markers for Dietary Composition Assessment aim to assess the accuracy of the determination method for plant intake by laying hens using n-alkanes as markers, while evaluating the recovery rates of the various n-alkanes in feces. The topic is original and falls within the topic of the journal.
The study is well designed. However, some minor corrections need to be made before being published.
Introduction:
- L 50-51: I suggest to add a comment regarding studies on poultry that involved surgical techniques, like precision-fed cecectomized rooster assay.
Materials and Methods
-L 84: in Table 1 you presented calculated nutrient content, not chemical analysis. Please correct the title of the table.
- L 84: please include in Table 1 the diet with 1 % alfalfa.
- L 117: How did you cooled the samples to 70°C?
- L 121: please mention the equipment used for drying the samples.
- L 122: please mention the type of the SPE column and the producer.
Author Response
We would like to express our sincere gratitude for your thorough review of our manuscript, titled "Quantitative and Qualitative Evaluation of Plant Intake in Laying Hens: n-Alkanes as Predictive Fecal Markers for Dietary Composition Assessment". Your insightful comments and suggestions have immensely contributed to the enhancement of the quality and clarity of our work.
- We thank you for recommending the addition of a reference related to surgical techniques in the introduction. We have incorporated the reference (Lorenz et al., 2013) to provide a comprehensive background about a surgical method to estimate feed intake from pasture in broilers and laying hens.
- Acknowledging the discrepancy in the title of Table 1, we will correct it to reflect that the table contains calculated nutrient content. Additionally, we have addressed your request to include the diet with 1% alfalfa in Table 1. However, in the Results section, we have included Table 2, where, at line 172, you will find the approximate composition of the diet with and without 1% alfalfa. This table plays a crucial role in illustrating the non-significant impact of the 1% alfalfa inclusion on the nutritional composition of the diet.
- We appreciate your queries regarding the cooling process, equipment used for drying, and details about the SPE column. Your suggestions have been implemented, and the relevant information is now provided in the manuscript.
- To achieve partial cooling from 90°C to 70°C, ice was added to the water bath.
- Reacti-Therm III #TS-18824 Heating Module in combination with nitrogen blowing on the surface to avoid oxidatition was used to dry the samples
- Column Type: Discovery® DSC-Si SPE Tube (500 mg, 3 mL). Producer: Supelco Inc.
Reviewer 2 Report
Comments and Suggestions for Authors
General Comments;
- Well written and thought-out paper. With cage-free being the upmost importance in the industry world along with welfare reasons, a study such as this one is important especially in countries that do allow their birds to forage outside on fields/grasslands. Looking into the specifics will allow for nutritionists to form reliable diets to ensure the birds are still maintaining and performing at the highest levels along with not having any negative health impacts.
- Things to consider for future research and I am not sure if it was looked at during this study.
o Were the birds weighed every time the average feed intake was recorded and what was the FCR mean and how did it compare with the commercial diet w/o alfalfa?
o Was performance looked at such as egg production and egg quality?
o Was there any mortality throughout the study and is there reasonable explanations for them?
- Considering the points above is critical when looking into the reasoning behind this study to begin with. As industry moves more and more to cage-free, factors to consider are the maintenance and performance of the birds and everything that might go into that such as the management and nutritional diets the birds are receiving.
- Lines 31-34; Please provide references for these two sentences/statements
- Lines 184-186; Why do you think that particular bird ate so much less vs the others?
Thank you!
Author Response
We would like to express our sincere gratitude for your thorough review of our manuscript, titled "Quantitative and Qualitative Evaluation of Plant Intake in Laying Hens: n-Alkanes as Predictive Fecal Markers for Dietary Composition Assessment". Your insightful comments and suggestions have immensely contributed to the enhancement of the quality and clarity of our work.
- We are grateful for your questions regarding bird weights, feed conversion ratio (FCR), and performance metrics. Birds were weighed specifically on the first (1426 ± 17.2 g) and last (1509 ± 15.1 g) days of the experiment. The feed conversion ratio (FCR) was calculated and was approximately 5.08 for the control diet and 4.97 for the mixed diet containing 1% alfalfa. The absence of significant difference helped us concluding the non-need for showing these details considering the short period of the trial.
- The current study has a focus on pre-laying phase, thus excluding metrics such as egg production and egg quality.
- Your concern about mortality has been addressed. We are pleased to report that no mortality occurred during the study, suggesting that the subjects adapted well to the conditions. We agree with your emphasis on considering maintenance and performance factors as the industry transitions to cage-free practices.
- References for statements in lines 31-34 and 184-186 have been provided as per your request.
https://info.bml.gv.at/en/topics/agriculture/agriculture-in-austria/animal-production-in-austria/egg-production-in-austria.html
- We appreciate your concern regarding the specific case of the excluded bird. The observed lower feed intake in this particular bird can be attributed to distinct behavior noted from the beginning of the experiment, characterized by reduced activity levels. It was apparent that this specific bird faced challenges adapting to the study conditions, resulting in a daily feed intake of approximately 42g. This was significantly below the group average of 70g, indicating a failure to successfully pass through the adaptation period.
Reviewer 3 Report
Comments and Suggestions for Authors
The manuscript presents a thoroughly designed study that is of very high relevance. However, there are two considerations in the methodology that need to be addressed by the authors in the manuscript.
1. Did you account for potential matrix effects of the feed or digesta during the extraction of alkanes?
2. The recovery of alkanes seems to be mainly based on assumptions about their composition stemming from the literature. My opinion is that more explanation about this approach is needed as all interested readers might not be fully familiar with such approaches. More importantly, wasn’t the recovery based on the total fecal collection? This part then needs to be further highlighted. Also, could the authors explain why a conventional digestibility marker was not used in this study?
L122: mention manufacturer of SPE column
Revise fig 1 with discernable captions etc. Now it looks like a screenshot.
Overall the discussion seems to be focused on the lower inclusion level of alfalfa compared to previous studies. If that is the main conclusion, then why was this study planned this way to begin with? I think that this part of the discussion can be more elaborate.
Author Response
We would like to express our sincere gratitude for your thorough review of our manuscript, titled "Quantitative and Qualitative Evaluation of Plant Intake in Laying Hens: n-Alkanes as Predictive Fecal Markers for Dietary Composition Assessment". Your insightful comments and suggestions have immensely contributed to the enhancement of the quality and clarity of our work.
- We appreciate the suggestion to consider matrix effects in the extraction of alkanes from feed or digesta, however, our study focused primarily on quantifying the recovery rate in a preliminary investigation. Although matrix effects were not addressed in this specific study, we acknowledge their potential significance. Future research may incorporate different designs to explore how the presence of other substances in the sample could impact the detection or quantification of n-alkanes, providing a more comprehensive understanding of the extraction process.
- In section L-87 of the materials and methods, we emphasized the utilization of total fecal collections. The study aimed to demonstrate the detectability of n-alkanes in diets, particularly those rich in ingredients like grains with very low concentrations of n-alkanes. This allowed us to estimate both the quality and quantity of feed intake. Other studies are in the pipeline, including the use of AIA as a marker for comparison. This section wasn't included in this manuscript due to its requirement for a distinct experimental design.
- In response to L122, the manufacturer of the SPE column used in the study is Supelco Inc. The specific column type employed was the Discovery® DSC-Si SPE Tube (500 mg, 3 mL). The provided details is now highlighted in the revised manuscript.
- The updated version of “Figure 1” now features improved resolution, ensuring a more comprehensive and clear presentation than the initial screenshot-like appearance.
- The study was strategically designed as one of the initial investigations utilizing n-alkanes in poultry. The primary objective was to validate the efficacy of this methodology on avian species, particularly considering the significantly lower inclusion levels of alfalfa (as an n-alkanes source) compared to established studies. This choice stemmed from the success of the method in ruminants, where forages constitute a substantial portion of daily intake and n-alkane concentrations are notably higher. The challenge in this study was to demonstrate the detectability of n-alkanes in feces even with minimal plant inclusion, allowing for accurate estimation of intake. As outlined in the manuscript, the study was structured to present a challenge by incorporating only 10% of the actual intake observed in prior studies that employed different methodologies.